# Two-Way Garment Transfer: Unified Diffusion Framework for Dressing and Undressing Synthesis

## Abstract

While recent advances in virtual try-on (VTON) have achieved realistic garment transfer to human subjects, its inverse task, virtual try-off (VTOFF), which aims to reconstruct canonical garment templates from dressed humans, remains critically underexplored and lacks systematic investigation. Existing works predominantly treat them as isolated tasks: VTON focuses on garment dressing while VTOFF addresses garment extraction, thereby neglecting their complementary symmetry. To bridge this fundamental gap, we propose the Two-Way Garment Transfer Model (TWGTM), to the best of our knowledge, the first unified framework for joint clothing-centric image synthesis that simultaneously resolves both mask-guided VTON and mask-free VTOFF through bidirectional feature disentanglement. Specifically, our framework employs dual-conditioned guidance from both latent and pixel spaces of reference images to seamlessly bridge the dual tasks. On the other hand, to resolve the inherent mask dependency asymmetry between mask-guided VTON and mask-free VTOFF, we devise a phased training paradigm that progressively bridges this modality gap. Extensive qualitative and quantitative experiments conducted across the DressCode and VITON-HD datasets validate the efficacy and competitive edge of our proposed approach.

## 1 Introduction

Computer vision has transformed fashion through virtual try-on (VTON) and try-off (VTOFF). VTON overlays clothes digitally for e-commerce, while VTOFF extracts garment designs for sustainability and AI recommendations. Despite their complementary functions, existing research treats them separately rather than as an integrated system.

Early VTON approaches (Ge et al., 2021; Han et al., 2019; 2018; He et al., 2022; Minar et al., 2020; Wang et al., 2018; Yang et al., 2020) predominantly employed Generative Adversarial Networks (Goodfellow et al., 2020) or other networks to implement a two-stage framework: First aligning garment patterns with human poses through specialized warping networks, then synthesizing realistic outputs via generators to integrate the deformed clothing with target personas. However, this paradigm faces inherent limitations in maintaining garment structural integrity, as imperfect geometric alignment in the warping stage frequently propagates distortions to final synthesis results. The emergence of diffusion models (Ho et al., 2020; Rombach et al., 2022; Song et al., 2022; Nichol & Dhariwal, 2021; Ramesh et al., 2021) has catalyzed new methodological directions in this field, diverging into two distinct research trajectories. One branch (Morelli et al., 2023; Gou et al., 2023; xujie zhang et al., 2023) enhances traditional pipelines by integrating diffusion models with preliminary warping predictions, leveraging their superior generation capabilities to refine deformation results. Conversely, alternative approaches (Zhu et al., 2023; Kim et al., 2024; Chong et al., 2025; Xu et al., 2025; Yang et al., 2024; Zhang et al., 2025) eliminate explicit warping operations entirely, instead employing diffusion models to autonomously learn spatial transformations through direct feature extraction from reference garments, thereby enabling implicit geometric reasoning during the generative process.

VTOFF, as an emerging field, faces key challenges in simultaneously resolving pose-induced deformations while maintaining clothing geometry, textures, and patterns. Current approaches (Velioglu

et al., 2024; Xarchakos & Koukopoulos, 2025) primarily employ diffusion models to implicitly learn inverse deformations, aligning with the second VTON paradigm.

Our key insight is that VTON and VTOFF constitute dual objectives within a unified deformation modeling paradigm. Specifically, VTON operates by estimating the forward deformation field to spatially align garments with target body poses, whereas VTOFF requires inference of the inverse deformation field to reconstruct canonical garment representations from pose-distorted inputs.

We analyze two representative methods: CatVTON (Chong et al., 2025) for VTON and TryoffAnyone (Xarchakos & Koukopoulos, 2025) for VTOFF, both minimizing parameters via input-level feature concatenation. As shown in Figue 1(a), their self-attention and FFN layers exhibit significant parameter similarity (6.4% at threshold 0.0005 → 65.6% at 0.05) by computing relative errors, indicating that minor architectural modifications suffice for joint task modeling.

Figure 1(b) displays a heatmap visualization of cross-task cosine similarity measurements for intermediate layer outputs at diffusion step T=30. This similarity, to a certain extent, arises because the reference features used in input concatenation are typically the processed target outcomes of the another task. This observation suggests that a simple transformation of the feature concatenation order in the latent space could be employed to alter the task objectives.

To realize this unified framework, we transcend conventional single-cue conditioning by establishing a dual-space guidance mechanism that synergizes latent-space feature alignment and pixel-space detail preservation. Specifically, latent space features are fused through spatial concatenation to maintain topological consistency of garment structures, while pixel space features are decomposed into complementary streams via a dual-branch architecture: a semantic abstraction module that distills category-aware garment semantics and a spatial refinement module that enhances fine-grained texture patterns. These disentangled features are then dynamically fused through an extended attention mechanism, which enables hierarchical feature integration across abstraction levels.

A fundamental distinction between VTON and VTOFF lies in mask utilization: while many VTON methods employ predefined inpainting masks, whereas VTOFF inherently lacks reliable mask guidance due to indeterminate clothing boundaries. To reconcile this spectrum of mask dependency, we develop a phased training protocol. Initially, we co-optimize high-level semantic feature extraction and a lightweight mask predictor in TaskFormer Module for VTOFF, using auxiliary loss on canonical garment shapes. Subsequently, we enable cross-task knowledge transfer by co-training all modules with task-specific attention gating, where mask conditioning is either provided (VTON) or predicted (VTOFF) in the Extended Attention Module. Through extensive experiments, we validate the efficacy of our design.

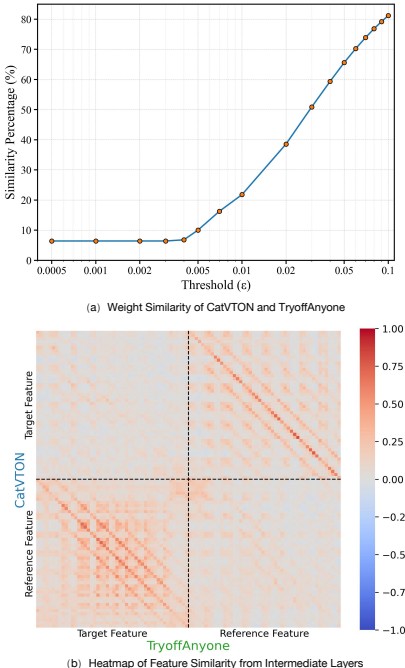

(a) Weight Similarity of CatVTON and TryoffAnyone

(b) Heatmap of Feature Similarity from Intermediate Layers

Figure 1: Analysis of the commonalities between VTON and VTOFF.

In summary, the primary contributions of our TWGTM can be highlighted as follows:

- We present, to the best of our knowledge, the first unified diffusion framework that achieves bidirectional garment manipulation through dual-path conditional guidance, where mask-guided VTON and mask-free VTOFF operations are bidirectionally derived through an implicit unified deformation field.

- We propose a dual-phase training strategy to address the task-specific mask discrepancy, simultaneously leveraging auxiliary segmentation loss for implicit canonical mask prediction in VTOFF.

- Extensive experiments validate our framework and training strategy, demonstrating the efficacy of jointly modeling these complementary virtual garment manipulation tasks.

## 2 RELATED WROK

### 2.1 VIRTUAL TRY-ON

Early methods(Han et al., 2018; Wang et al., 2018; Ge et al., 2021; Xie et al., 2023) established a two-stage pipeline: first aligning garments to target poses through geometric transformations (e.g., Thin Plate Spline(TPS) warping(Yang et al., 2020; Minar et al., 2020; Duchon, 1977; Lee et al., 2019), flow estimation(Bai et al., 2022; Li et al., 2021; Chopra et al., 2021; Zhou et al., 2016) or landmark(Yan et al., 2023; Liu et al., 2021a; Chen et al., 2023; Xie et al., 2020)), then synthesizing realistic try-on results using GANs or related architectures. But they easily suffered from error propagation due to imperfect warping, artifacts at garment-person boundaries, and heavy reliance on auxiliary inputs.

The rise of diffusion models catalyzed two distinct research trajectories: (1) Warping-enhanced diffusion frameworks(Morelli et al., 2023; Gou et al., 2023; xujie zhang et al., 2023) integrate preliminary warping predictions with diffusion models to refine alignment errors and improve texture fidelity. These hybrid methods leverage diffusion's generative strength to correct distortions while retaining structural priors from warping. (2) Warping-free diffusion frameworks(Zhu et al., 2023; Kim et al., 2024; Xu et al., 2025) eliminate explicit geometric alignment entirely, instead training diffusion models to implicitly infer spatial transformations through direct garment feature extraction. By encoding garments and employing attention mechanisms, these approaches autonomously learn deformation rules, enabling flexible handling of complex poses and non-rigid fabrics. Recent advancements like MMTryon(Zhang et al., 2024) further reduce dependency on auxiliary inputs by incorporating textual guidance and multi-modal conditioning, broadening usability for arbitrary garment-person pairs.

### 2.2 VIRTUAL TRY-OFF

VTOFF has recently emerged as a novel research direction in fashion-oriented computer vision, aiming to reconstruct canonical garment images from dressed human photos. Two pioneering works demonstrate promising approaches: TryOffDiff(Velioglu et al., 2024) achieves precise segmentation of target garment regions through its SigLIP-conditioned latent diffusion framework, yet encounters persistent limitations in reconstructing fine-grained details (e.g., embroidery patterns) and maintaining color fidelity across varying illumination conditions. Subsequently, TryOffAnyone achieves computationally efficient garment reconstruction via a mask-integrated StableDiffusion variant with transformer tuning, but suffers from spatial inaccuracies (e.g., over-inference at sleeve joints, or distortions in adjacent garments).

## 3 METHODOLOGY

### 3.1 PRELIMINARY

Latent Diffusion Model(Rombach et al., 2022) establishes a hierarchical framework for conditional image generation through latent space manipulation. The architecture comprises three core components: (1) A CLIP text encoder(Radford et al., 2021) $\mathcal{E}_T$ that projects prompts $y$ into a 768-dimensional embedding space, (2) A variational autoencoder (VAE)(Kingma et al., 2013) with encoder $\mathcal{E}$ compressing input images $I \in R^{3 \times H \times W}$ into lower-dimensional latent representations $z_0 = \mathcal{E}(I) \in R^{c \times h \times w}$ (typically $h = \frac{H}{8}, w = \frac{W}{8}, c = 4$), together with a decoder $\mathcal{D}$, and (3) A time-conditional U-Net(Ronneberger et al., 2015) $\epsilon_\theta$ that progressively denoises corrupted latents $z_t$ over $T$ diffusion steps. The forward process follows the Markov chain:

$$\alpha_t := \prod_{s=1}^{t}(1 - \beta_s), z_t = \sqrt{\alpha_t}z_0 + \sqrt{1 - \alpha_t}\epsilon, \tag{1}$$

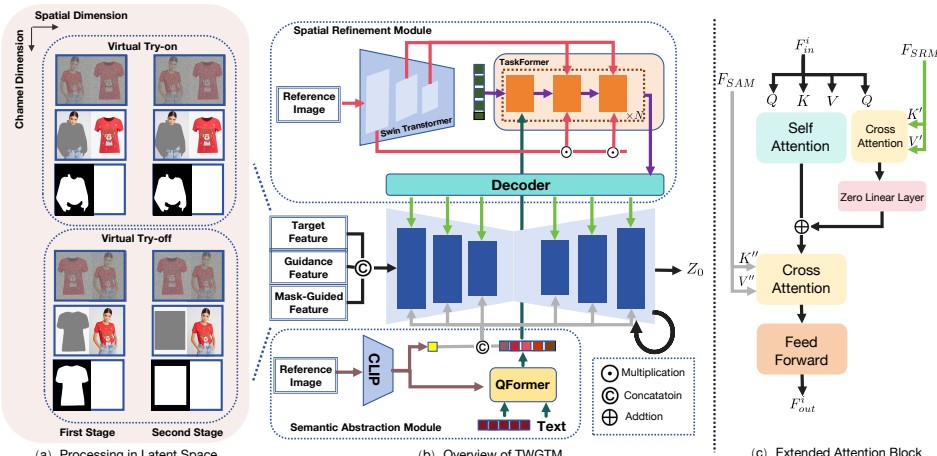

Figure 2: (a) The illustration shows latent space input processing in the first and second stages of training. (b) Overview of TWGTM, omitting the VAE encoder and decoder. (c) The Extended Attention Module integrates outputs from the Spatial Refinement Module and Semantic Abstraction Module through dual cross-attention operations.

where $\beta_s$ defines a noise schedule from $\beta_1$ to $\beta_T$, $\epsilon \sim \mathcal{N}(0, \mathbf{I})$. The model optimizes the noise prediction objective:

$$\mathcal{L}_{\text{DM}} = \mathbb{E}_{\mathcal{E}(I), \epsilon \in \mathcal{N}(0,1), t, y} \left[ \| \epsilon - \epsilon_\theta(z_t, t, \mathcal{E}_T(y)) \|_2^2 \right]. \tag{2}$$

## 3.2 PROCESSING IN LATENT SPACE

**Training Phase.** Figure 2(a) illustrates the feature concatenation process during training. For VTON, we spatially concatenate the model image $x$ and flattened garment $c$ to form the target feature $h_i = [x, c] \in R^{3 \times H \times 2W}$. Let $M$ denote the localized garment mask and $x_a$ represent the masked person image, where $x_a$ is obtained by: $x_a = (1 - M) \otimes x$, with $\otimes$ explicitly defined as element-wise multiplication. We further concatenate the mask-guided feature $h_m = [1 - M, ones(M)] \in R^{1 \times H \times 2W}$ (where $ones(M)$ is an all-ones mask with identical dimensions to $M$) and the guidance feature $h_f = [x_a, c] \in R^{3 \times H \times 2W}$. The variables $h_i$ and $h_f$ are subsequently encoded into the latent space through the VAE encoder $\mathcal{E}$, while $h_i$ is perturbed by additive Gaussian noise scaled according to the diffusion timestep t and $h_m$ is resized to match the spatial dimensions of the latent space representation. The final input tensor is constructed via channel-wise concatenation: $I_t = [Noise_t(\mathcal{E}(h_i)), \mathcal{E}(h_f), resize(h_m)] \in R^{(4+4+1) \times \frac{H}{8} \times \frac{2W}{8}}$.

For VTOFF, this process reverses the spatial order between $x$ and $c$, along with their corresponding mask-guided feature and guidance feature. Notably, in the first training stage, the garment mask is directly synthesized from the conditional input $c$, focusing on developing the model's inpainting capability for precisely reconstructing designated garment regions. In contrast, the second stage replaces these masks with morphologically augmented square-shaped masks (generated via erosion-dilation operations), deliberately challenging the model's geometric reasoning capacity to recover accurate shape boundaries.

**Inference Phase.** We retain both the guidance feature and mask-guided feature while replacing the initial input channels with a fully noised tensor. Especially, for VTOFF, the original garment mask in $h_m$ is substituted with an all-zeros mask $zeros(M)$ matching $M$'s dimensions and the guidance feature undergo corresponding adjustments. Additionally, rectangular masks (e.g., bounding box) can serve as an alternative for garment proportion optimization.

### 3.3 PROCESSING IN PIXEL SPACE

**Semantic Abstraction Module.** This module combines a CLIP image encoder with QFormer(Li et al., 2023b), where the CLIP encoder remains frozen throughout training. The reference image is first encoded by CLIP to obtain the global semantic feature $F_{CLIP\_G} \in R^{B \times 1 \times D}$, where the final-layer token sequence $F_{CLIP\_S} \in R^{B \times L \times D}$ serves as the input to QFormer. Context-aware filtering is implemented by conditioning QFormer with semantically-grounded text prompts (e.g., "upper garment" for torso clothing), which enables targeted feature selection from CLIP's outputs. This cross-modal interaction process can be formulated as:

$$F_{QF} = \text{QFormer}\,(F_{CLIP\_S},\, T_p) \in \mathbb{R}^{B \times N \times D}, \tag{3}$$

where $T_p$ represents the encoded text features and N denotes the number of learnable query tokens. The final module output is formed by concatenating CLIP's global semantic feature $F_{CLIP\_G}$ with QFormer's filtered representations $F_{QF}$ along the sequence dimension, resulting in a composite sequence

$$F_{SAM} = [F_{CLIP\_G}, F_{QF}] \in R^{B \times (1+N) \times D}. \tag{4}$$

**Spatial Refinement Module.** This module employs distinct query embeddings to learn region-specific features across multi-scale inputs, enabling precise spatial detail extraction from reference images. We initially employ Swin Transformer (Liu et al., 2021b) as the image encoder to extract multi-scale features $H = [H_1, H_2, H_3]$ (1/4, 1/8, 1/16 resolutions) from the reference image. These multi-resolution features, alongside the semantic features from QFormer outputs $F_{QF}$ as inputs, are jointly fed into TaskFormer's three scale-specific processing blocks, which are based on a transformer-decoder-like architecture (Vaswani et al., 2017). This architecture independently processes each scale while facilitating cross-granularity interaction between spatial and semantic features.

TaskFormer employs different learnable queries to dynamically attend to distinct spatial regions across hierarchical feature maps. Each processing unit comprises three blocks operating on individual feature scales ($F_{QF}, H_2, H_3$), where the conventional cross-attention is replaced by masked attention layers and preceded self-attention layers (following Mask2Former (Cheng et al., 2022)) to enforce spatial coherence through generative region masks. The i-th processing unit computes $F_i = H'_3$, where features are refined sequentially through cascaded blocks:

$$F'_{QF} = \text{Block}_1(F_{i-1}, F_{QF}, M^1_{i-1}), \tag{5}$$

$$H'_2 = \text{Block}_2(F'_{QF}, H_2, M^2_{i-1}), \tag{6}$$

$$H'_3 = \text{Block}_3(H'_2, H_3, M^2_{i-1}), \tag{7}$$

with hierarchical masks $M^1_{i-1}$ and $M^2_{i-1}$ propagated from the (i-1)-th unit to enforce task-specific spatial constraints.

Subsequently, the dual projection branches process each unit's outputs through distinct operations. The mask-space projection applies multilayer perceptron (MLP) and linear layer to both spatially and semantically constrained attention masks

$$M^1_i = \sigma\,(\text{Linear}(F_i) \odot F_{QF}) \in [0, 1]^{B \times K \times N}, \tag{8}$$

$$M^2_i = \sigma\,(\text{MLP}_{\text{mask}}(F_i) \odot H_1) \in [0, 1]^{B \times K \times H \times W}, \tag{9}$$

where $\sigma$ is the sigmoid function, $\odot$ denotes tensor multiplication, $H \times W$ is the spatial resolution of the feature map, $K$ denotes the number of learnable query tokens. These masks dynamically highlight task-critical regions (e.g., garment patterns in virtual try-on) to enhance details in different regions. Simultaneously, the task-space projection employs query-specific processing, where the first query explicitly predicts flattened garment masks for VTOFF via

$$TFQ_0 = \sigma(MLP^0_{task}(F_i) \odot H_1) \in [0, 1]^{B \times 1 \times H \times W}, \tag{10}$$

while subsequent queries ($j \geq 1$) implicitly enhance diffusion-generated details through unconstrained refinement

$$TFQ_j = MLP^j_{task}(F_i) \odot H_1 \in R^{B \times (K-1) \times H \times W}. \tag{11}$$

The TaskFormer eventually outputs three-scale features with the same resolution as the Unet's latent space.

Subsequently, the lightweight Decoder combines convolutional layers and basic transformer blocks to project TaskFormer's outputs into UNet-compatible feature dimensions, preserving critical spatial information while ensuring computational efficiency. This process is formulated as:

$$F_{SRM} = Decoder(TFQ). \tag{12}$$

**Extended Attention Block.** As illustrated in Figure 2(c), this module integrates spatial features $F_{SRM}$ with self-attention learned features from UNet to compensate for information loss in latent space and enhance detailed image generation. The proposed Zero Linear Layer serves dual purposes: (1) gradually introduces spatial feature influences to reduce learning difficulty, and (2) suppresses noise in spatial features while amplifying discriminative patterns. Specifically, UNet features are decomposed into query $(Q)$, key $(K)$, and value $(V)$ tensors. These first undergo standard self-attention: $SelfAttn(Q, K, V) = softmax(\frac{QK^T}{\sqrt{d}})V$. Simultaneously, cross-attention is performed between $Q$ and spatial-derived $K', V'$: $CrossAttn(Q, K', V') = softmax(\frac{QK'^T}{\sqrt{d}})V'$. The cross-attention outputs are modulated through the Zero Linear Layer $(ZLL)$ before element-wise summation with self-attention features:

$$F_{fused} = SelfAttn + ZLL(CrossAttn). \tag{13}$$

Finally, the additional cross-attention between the fused features $F_{fused}$ and the high-level semantic features $F_{SAM}$ serves to extract complementary information, thereby enhancing the overall feature representation.

## 3.4 TRAINING STRATEGY

Inpainting-based VTON utilizes segmentation-based methods to explicitly locate garment replacement regions, while VTOFF requires simultaneous shape inference and texture inpainting due to indeterminate damage regions after virtual garment removal.

To balance the inherent difficulty disparity between tasks, we implement a phased training strategy. Stage 1 focuses on inpainting capability by using the generated flattened garment mask as the VTOFF inpainting region. The model predicts flattened garment masks through spatial features from the first query vector of the TaskFormer, while reference image features are exclusively integrated via UNet's native cross-attention blocks. The training objective for the first stage is as follows:

$$\mathcal{L} = \mathbb{E} \left\| \epsilon - \epsilon_\theta(I_t, t, \tau_{SAM}(\mathbf{x}_{ref}) \right\|_2^2 + \lambda \mathcal{L}_{mask}, \tag{14}$$

$$\mathcal{L}_{mask} = \lambda' \mathcal{L}_{dice} + \lambda'' \mathcal{L}_{bce}. \tag{15}$$

Stage 2 enhances shape awareness by applying morphological operations (erosion and dilation) to generate square inpainting masks. This phase combines spatial features from the Spatial Refinement Module with self-attention learned features to reinforce garment shape constraints, with the Extended Attention Block activated for feature fusion. The training objective for the second stage is as follows:

$$\mathcal{L} = \mathbb{E} \left\| \epsilon - \epsilon_\theta(I_t, t, \tau_{SAM}(\mathbf{x}_{ref}), \tau_{SRM}(\mathbf{x}_{ref}) \right\|_2^2. \tag{16}$$

## 4 EXPERIMENTS

### 4.1 DATASETS AND METRICS

We conduct the experiments using two publicly available datasets, VITON-HD(Choi et al., 2021) and DressCode(Morelli et al., 2022). The VITON-HD dataset comprises over 10,000 pairs of upper-body garments, while the DressCode dataset encompasses three categories of clothing: upper garments, lower garments, and dresses, with a total of over 40,000 image pairs.

Following previous works, the evaluation employs SSIM (and variants)(Wang et al., 2004; Tang et al., 2011) for structural accuracy, LPIPS(Zhang et al., 2018) and DISTS(Ding et al., 2020) for texture fidelity, FID(Heusel et al., 2017) and KID(Bińkowski et al., 2021) for perceptual realism, and DINO(Zhang et al., 2022) similarity and CLIP-FID for semantic alignment. More details can be found in the appendix.

Figure 3: Qualitative comparison of VTON and VTOFF results with baselines.

## 4.2 QUANTITATIVE COMPARISON

Table 1: Quantitative comparison of VTON results with baselines on the DressCode dataset. The best and suboptimal results are demonstrated in bold and underlined, respectively.

| Method | Upper Body | | | | Lower Body | | | | Dresses | | | |
|---|---|---|---|---|---|---|---|---|---|---|---|---|
| | FID↓ | DINO↑ | SSIM↑ | LPIPS↓ | FID↓ | DINO↑ | SSIM↑ | LPIPS↓ | FID↓ | DINO↑ | SSIM↑ | LPIPS↓ |
| GP-VTON | 17.585 | 0.864 | 0.779 | 0.200 | 21.411 | 0.904 | 0.771 | 0.206 | 13.816 | 0.893 | 0.794 | 0.156 |
| LaDI-VTON | 14.108 | 0.883 | 0.919 | 0.055 | 14.215 | 0.926 | 0.914 | 0.058 | 16.548 | 0.859 | 0.863 | 0.077 |
| IDM-VTON | **7.277** | **0.941** | 0.929 | **0.033** | 8.313 | **0.967** | 0.913 | **0.035** | 9.018 | 0.921 | **0.884** | 0.075 |
| CatVTON | 7.805 | 0.919 | 0.929 | 0.034 | 8.910 | 0.937 | 0.912 | 0.045 | 8.890 | 0.906 | 0.865 | **0.062** |
| Ours | 7.497 | 0.939 | **0.941** | 0.043 | **8.298** | 0.960 | **0.922** | 0.054 | **8.412** | **0.923** | 0.881 | **0.062** |

For VTON, Table 1 and Table 2 display quantitative comparisons of TWGTM (ours) with other state-of-the-art methods on the VITON-HD and DressCode test datasets, respectively. Our method demonstrates competitive performance on both datasets, achieving advanced levels of performance across most metrics. On the Dress-Code dataset, some metrics fall short of the expected or desired outcomes, which we speculate primarily stem from inherent color and texture inconsistencies between the model and the flattened garment images. These inconsistencies are likely caused by variations in lighting conditions during the original photography process. Further analysis can be found in the appendix.

Table 2: Quantitative comparison of VTON results with baselines on the VITON-HD dataset.

| Method | VITON-HD | | | |
|---|---|---|---|---|
| | FID↓ | DINO↑ | SSIM↑ | LPIPS↓ |
| DCI-VTON | 7.119 | 0.940 | 0.881 | 0.065 |
| MV-VTON | 8.597 | 0.942 | 0.887 | 0.060 |
| GP-VTON | 8.939 | 0.899 | 0.880 | 0.068 |
| LaDI-VTON | 11.297 | 0.924 | 0.869 | 0.075 |
| IDM-VTON | 6.098 | 0.957 | 0.865 | 0.074 |
| CatVTON | **5.693** | 0.954 | 0.871 | 0.060 |
| Ours | 6.107 | **0.960** | **0.905** | **0.055** |

For VTOFF, our method in Table 3 demonstrates state-of-the-art performance on the VITON-HD dataset, significantly outperforming existing approaches across most metrics. Compared to others, our method achieves the lowest DISTS score, indicating superior preservation of structural and textural fidelity, which aligns with its designation as the primary metric for VTOFF.

## 4.3 QUALITATIVE COMPARISON

For VTON, our proposed model demonstrates superior capability in preserving fine-grained details and addressing color discrepancies from reference images. As shown in the first row of Figure 3(a)

Table 3: Quantitative comparison of VTOFF results with baselines on the VITON-HD dataset.

| Method | SSIM↑ | MS-SSIM↑ | CW-SSIM↑ | LPIPS↓ | FID↓ | CLIP-FID↓ | KID↓ | DISTS↓ |
|---|---|---|---|---|---|---|---|---|
| TryOffDiff | **0.727** | 0.526 | 0.422 | 0.414 | 21.397 | 8.627 | 7.6 | 0.246 |
| TryOffAnyone | 0.723 | 0.583 | 0.513 | 0.340 | 11.553 | **5.131** | 2.0 | 0.213 |
| Ours | 0.721 | **0.590** | **0.524** | **0.332** | **10.393** | 5.651 | **1.5** | **0.195** |

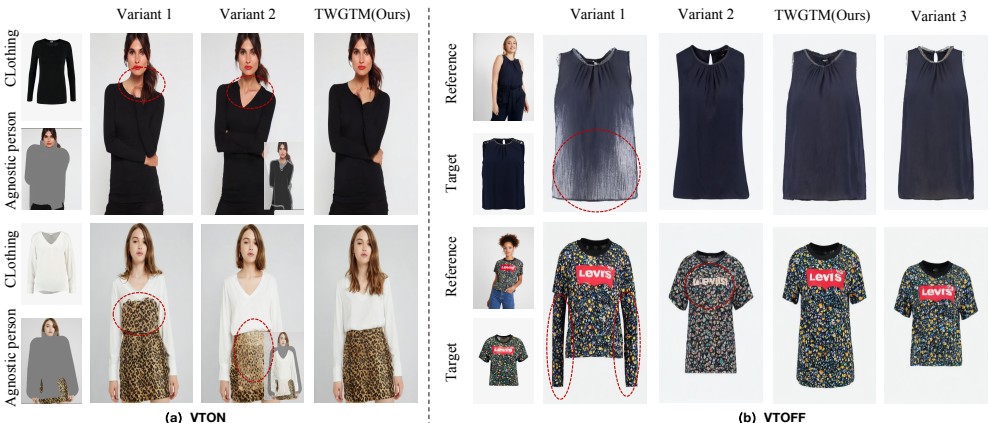

Figure 4: Generated outcomes of ablation variants in both VTON and VTOFF tasks.

, the gray "ALL" lettering on black clothing exhibits reduced clarity due to low contrast, yet our synthesized garment retains these subtle texture details with fidelity. Furthermore, the second-row experimental results demonstrate precise preservation of textual elements under non-uniform color distributions, underscoring our framework's robustness in handling complex garment textures.

For VTOFF, as shown in Figure 3(b), TryOffDiff struggles to preserve intricate patterns in complex garments, introducing color distortions or erroneous inferences, while TryOffAnyOne partially addresses detail loss but easily suffers from feature redundancy, erroneously incorporating extraneous garment characteristics. In comparison, our model better preserves the color, texture, and shape of the garments.

### 4.4 ABLATION STUDIES

Compared to TWGTM, **Variant 1** (removing SRM and relying solely on SAM via standard attention mechanism) and **Variant 8** (removing QFormer while maintaining the outputs from the CLIP model) in Table 4 exhibits degraded performance across key evaluation metrics for both VTON and VTOFF. As shown in Figures 4, **Variant 1** exhibits detail loss (e.g., texture distortion in the hand region and partial color deviations), along with localization issues in inpainting regions (e.g., excessive inference on skirts caused by failure in agnostic regions and the generation of extraneous sleeves).

We also evaluated two alternative feature fusion methods: **Variant 6** (SRM output concatenation before self-attention) and **Variant 7** (ip-Adapter integration). As shown in Table 4, both variants showed degraded performance, with more severe degradation in VTOFF.

**Variant 2** (replacing spatial concatenation with deformed garment fusion through reference image-warped garment combination) achieves marginally higher SSIM scores, but reveals artifacts induced by garment deformation. Specifically, spatial ambiguity induced by inaccurate warping disrupts garment-body alignment. Issues such as pose changes and faded clothing colors caused by erroneous warping in VTON from Figure 4(a), as well as the loss of textual details in VTOFF from Figure 4(b), also corroborate this problem. This conclusively demonstrates that spatial concatenation in latent space remains essential for preserving positional coherence between garments and body regions.

Table 4: Ablation studies of network components in our model.

| Setting | Virtual Try-on | | | | Virtual Try-off | | | |
|---|---|---|---|---|---|---|---|---|
| | FID↓ | DINO↑ | SSIM↑ | LPIPS↓ | FID↓ | SSIM↑ | LPIPS↓ | DISTS↓ |
| Variant 1(w/o Spatial Refinement Module) | 6.317 | 0.957 | 0.891 | 0.062 | 11.748 | 0.690 | 0.370 | 0.211 |
| Variant 2(w/o spatial concat) | 6.604 | 0.950 | **0.906** | 0.056 | 15.288 | 0.738 | 0.309 | 0.208 |
| Variant 3(w Mask2BBox) | - | - | - | - | **9.201** | **0.769** | **0.225** | **0.174** |
| Variant 4 (Scratch-Only VTON Training) | 6.165 | **0.960** | 0.905 | **0.055** | - | - | - | - |
| Variant 5 (Scratch-Only VTOFF Training) | - | - | - | - | 13.398 | 0.680 | 0.367 | 0.217 |
| Variant 6 (Pre-SelfAttention Feature Concatenation) | 6.130 | 0.959 | 0.904 | **0.055** | 15.082 | 0.629 | 0.448 | 0.241 |
| Variant 7 (w ip-Adapter) | 6.168 | 0.959 | 0.904 | 0.056 | 14.182 | 0.651 | 0.420 | 0.230 |
| Variant 8 (w/o QFormer) | 6.183 | **0.960** | 0.904 | 0.056 | 14.094 | 0.668 | 0.386 | 0.225 |
| TWGTM(Ours) | **6.107** | **0.960** | 0.905 | **0.055** | 10.393 | 0.721 | 0.332 | 0.195 |

**Variant 3** maintains the training-time configuration for VTOFF inference, using an enhanced rectangular inpainting region to guide synthesis. It shows significant metric improvements, proving that explicit modeling of garment geometry (aspect ratio, position) boosts both generation quality and controllable customization. As demonstrated in Figure 4(b), the predefined rectangular inpainting region effectively regulate the proportion of flat clothing in the generated results.

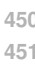
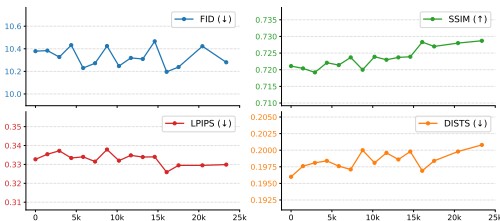

Figure 5: The impact of fine-tuning for VTON on the performance of VTOFF.

Figure 6: The impact of fine-tuning for VTOFF on the performance of VTON.

**Variant 4** and **Variant 5** are task-specific models trained from scratch. As shown in Table 4, VTON achieves stronger performance due to mask-guided restoration, while VTOFF shows greater performance variance. Simultaneously, we conducted fine-tuning on each of the two single tasks individually to explore the impact of fine-tuning one task on the performance of the other. As shown in Figure 5, when VTON was fine-tuned in isolation, there was a partial improvement observed in the FID, SSIM, and LPIPS metrics for VTOFF. Similarly, Figure 6 demonstrates a comparable improvement in the FID and LPIPS metrics for VTON. These results substantiate our hypothesis that, to a certain extent, these two tasks can mutually enhance each other's performance.

## 5 Conclusion

In this work, we present a unified diffusion framework Two-Way Garment Transfer Model (TWGTM) for bidirectional virtual garment manipulation, addressing both mask-guided VTON and mask-free VTOFF through reversible spatial transformations. Our model innovatively bridges the gap between these two complementary tasks by leveraging dual-conditioned guidance from both latent and pixel spaces. Latent features are fused via spatial concatenation to maintain structural integrity, while pixel features are refined through dedicated modules for semantic abstraction and spatial detail enhancement. The Extended Attention Block then integrates these features effectively. Additionally, we introduce a phased training strategy to mitigate the mask dependency issue between VTON and VTOFF. Extensive experiments demonstrate the superior performance of our method.

## 6 Ethics Statement

This research strictly follows the ICLR Code of Ethics. No studies involving human participants or animal subjects were conducted. All datasets—including **VITON-HD and DressCode**—were

obtained in accordance with applicable usage policies, ensuring full respect for privacy. We actively guarded against any form of bias or discriminatory results, did not utilise personally identifiable information, and avoided experiments that could compromise privacy or security. Transparency and integrity were maintained throughout the investigation.

## 7 REPRODUCIBILITY STATEMENT

To promote reproducibility, we have released our code and data anonymously and documented the full pipeline—training procedures, model hyper-parameters, and hardware setup—in the paper. These resources should allow the community to replicate our findings and build upon them.

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

# A APPENDIX

## A.1 LLM USAGE

We used a large language model (LLM) exclusively to polish the language—improving grammar, clarity, and flow. The model did not contribute to research ideas, methodology, or data analysis. All scientific content remains the authors' responsibility, and we have verified that the LLM-assisted text meets ethical standards.

## A.2 IMPLEMENTATION DETAILS

We use Paint by Example(Yang et al., 2023) as the backbone of our method and the swin transformer is a standard Swin-B(Liu et al., 2021b) pretrained on ImageNet(Deng et al., 2009). Additionally, the weights of QFormer(Li et al., 2023b) are initialized from BLIP-Diffusion(Li et al., 2023a), while those of taskFormer are partially initialized from Mask2Former(Cheng et al., 2022).

The hyper-parameter $\lambda$ is set to 5e-2, $\lambda'$ is set to 0.9, and $\lambda''$ is set to 0.1. We employ the AdamW optimizer(Loshchilov & Hutter, 2019), with an initial learning rate of 1e-5 that increases linearly from 0 during the first 1,000 warmup steps. In QFormer, the number of learnable query tokens is set to N = 32, while TaskFormer extends this capacity to K = 100 learnable query tokens to handle complex task hierarchies.

## A.3 Metrics

We assess our VTON ability using three categories of metrics: (1) Reconstruction with Structural Similarity Index Measure (SSIM)(Wang et al., 2004) and Learned Perceptual Image Patch Similarity (LPIPS)(Zhang et al., 2018) to measure pixel-level alignment and texture fidelity; (2) Perceptual Quality via Fréchet Inception Distance (FID)(Heusel et al., 2017) to evaluate realism on VITON-HD and DressCode datasets; and (3) Semantic Consistency through DINO similarity(Zhang et al., 2022) for high-level feature matching. For VTOFF, we follow the existing SOTA garment reconstruction method TryOffDiff, combining structural accuracy (SSIM variants(Wang et al., 2004; Tang et al., 2011) for pixel alignment) with multi-level perceptual assessment (LPIPS and Deep Image Structure and Texture Similarity (DISTS)(Ding et al., 2020) for texture, FID and Kernel Inception Distance (KID)(Bińkowski et al., 2021) for realism, CLIP-FID for semantic alignment).

## A.4 Dataset Analysis

As illustrated in the first row of Figure 7, the absence of distinctive visual features in the model's upper garment makes it challenging to classify the item as either a regular short-sleeved top or a one-piece outfit, resulting in inherent ambiguity for the model. Such cases typically require supplementary feature cues (e.g., textual context) to mitigate misclassification.

In the second row, significant color disparities between the worn garments and their flat-laid counterparts are observed, primarily attributable to environmental variations during photography. These inconsistencies substantially exacerbate the difficulty of bidirectional garment style transfer tasks.

Notably, in the final sample of the second row, while the target garment is an upper-body piece, the model exhibits a tendency to misinterpret it as a skirt due to the visually blurred boundary between the top and skirt in the model's pose. This perceptual ambiguity persists even in human visual cognition, underscoring the intrinsic challenges in fine-grained garment segmentation under such conditions.

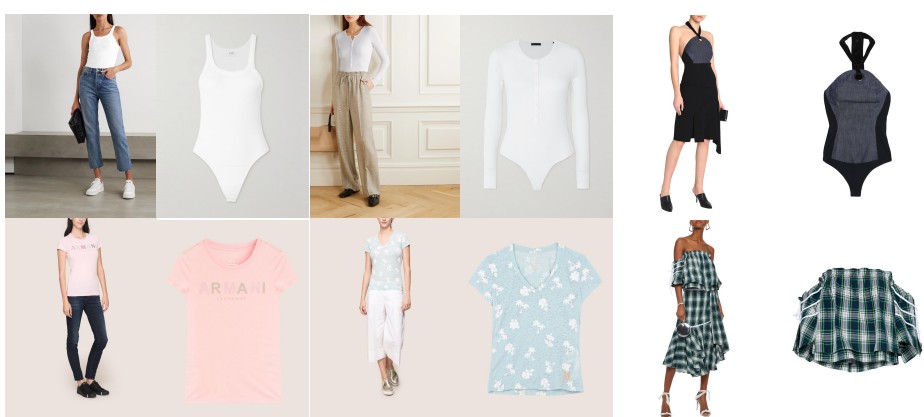

Figure 7: Data features of DressCode dataset.

## A.5 Additional Results

### A.5.1 Analysis of Failure Cases.

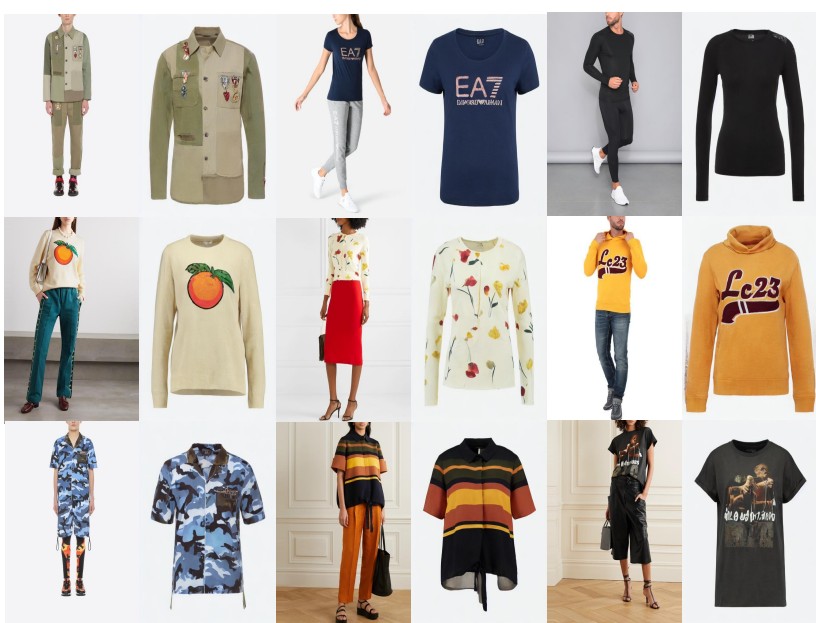

Figure 8: VTOFF results of our model trained on VITON-HD dataset, tested on DressCode dataset.

The DressCode dataset, characterized by a predominantly white background, presents certain challenges when processing garments of extreme colors, particularly white, which can lead to noticeable color distortions. As illustrated in the first row of Figure 9, white garments tend to exhibit heightened contrast against the background. Furthermore, when dealing with black garments, there is an occasional occurrence of color fading, as depicted in the second row. The presence of prominent accessories (e.g., the belt shown in row 3) may cause the model to misclassify them as intrinsic garment features, resulting in distorted representations. Furthermore, specular highlights caused by smooth material surfaces under studio lighting (as seen in row 4) are persistently replicated in the generated outputs, introducing unintended material rendering artifacts.

### A.5.2 CROSS-DATASET EVALUATION.

We trained our model on the VITON-HD dataset and conducted inference on the Dress-Code dataset to evaluate its generalization capability to unseen data. The results, as shown in figure 8, demonstrate that our model maintains positive outcomes in the virtual try-off task, effectively generating plausible outputs despite the domain shift. However, fine-grained reasoning errors persist, particularly in garment

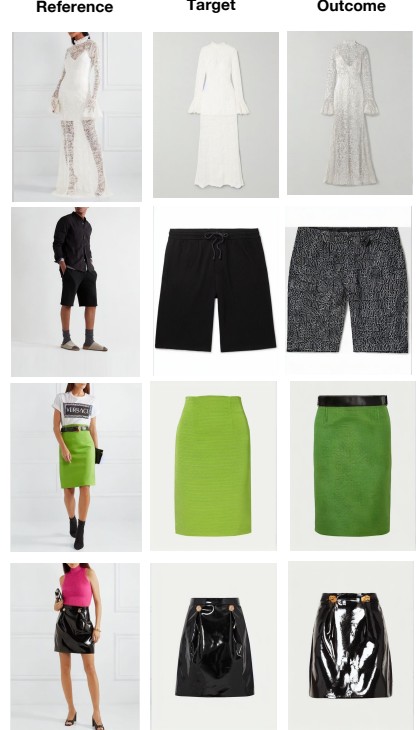

Figure 9: Illustrative failure cases.

hem inference. As demonstrated by the first two examples in the bottom row of Figure 8, these discrepancies primarily stem from inter-dataset distributional shifts. These observations underscore both the robustness and current limitations of our approach in handling diverse and unseen data distributions.

Additional results are displayed in Figure 10.

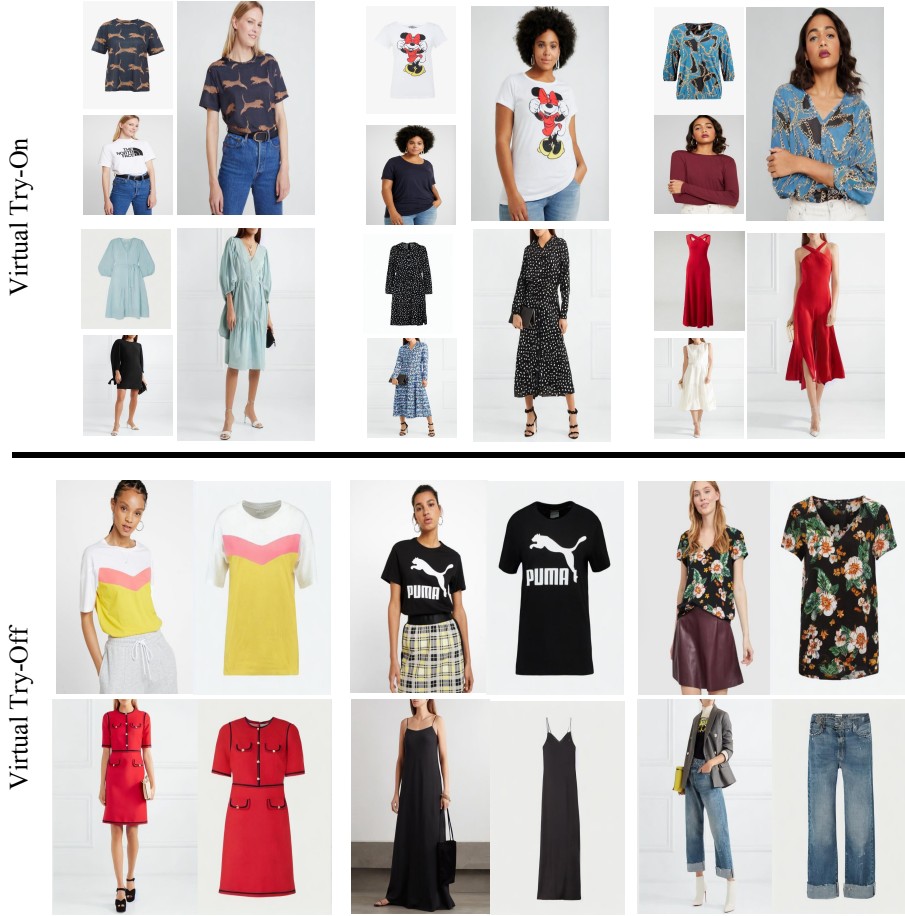

Figure 10: More examples of experimental results.

