# OpenReview forum: "Two-Way Garment Transfer: Unified Diffusion Framework for Dressing and Undressing Synthesis"
_ICLR.cc/2026/Conference — Submitted to ICLR 2026_

### Official Review · Reviewer_9F5u · 2025-10-19

**Soundness:** 3
**Presentation:** 3
**Contribution:** 3
**Rating:** 4
**Confidence:** 5

**Summary:**

This paper proposes a unified framework for TryOn and TryOff, implementing both TryOn and TryOff through dual-conditional control. It achieves excellent performance on both TryOn and TryOff tasks.

**Strengths:**

1. As a unified framework for Tryon and Tryoff, it has value in promoting community development
2. The mask-free form greatly facilitates the application of the method
3. Conducted extensive and thorough experiments

**Weaknesses:**

1 . The overall approach is similar to MagicCloth and ControlNet.
2. The method design is too complicated, and it seems that not all modules are designed meaningfully.
3. Lack of comparison with advanced DiT-based methods such as FiTDiT
4. In the results of Figure 8, the color of the clothes in the second row is obviously faded, indicating that the method is not robust enough.

**Questions:**

see weakness

---

### Official Review · Reviewer_1WAK · 2025-10-28

**Soundness:** 2
**Presentation:** 2
**Contribution:** 2
**Rating:** 4
**Confidence:** 3

**Summary:**

This paper addresses the underexplored problem of virtual try-off (VTOFF) by proposing the Two-Way Garment Transfer Model (TWGTM), a unified framework for both virtual try-on (VTON) and VTOFF tasks. The model leverages bidirectional feature disentanglement and dual-conditioned guidance from latent and pixel spaces to jointly synthesize clothing-centric images. A phased training paradigm is introduced to bridge the asymmetry between mask-guided VTON and mask-free VTOFF. Extensive experiments on DressCode and VITON-HD datasets demonstrate the effectiveness and competitiveness of the proposed approach.

**Strengths:**

- This paper is well-written and is easy to follow.
- The experiments demonstrate the effectiveness and competitive performance of the proposed method.

**Weaknesses:**

- The parameter similarity comparison in Figure 1a is insufficiently substantiated; there is no clear standard for what constitutes a sufficiently high similarity, and at minimum, a comparison with the base model parameters should be provided.
- The definition and role of the reference image are unclear.
- The performance improvements achieved by the proposed method are marginal.
- The motivation behind the design of the Spatial Refinement Module, as well as the reasoning for each component and operation within it, is not adequately explained.
- The Semantic Abstraction Module appears to closely resemble the IP-Adapter, lacking sufficient distinction or novelty.

**Questions:**

See Weaknesses.

---

### Official Review · Reviewer_sauk · 2025-10-30

**Soundness:** 3
**Presentation:** 2
**Contribution:** 2
**Rating:** 2
**Confidence:** 5

**Summary:**

The paper presents TWGTM, the first unified diffusion framework for bidirectional garment transfer. It jointly handles masked virtual try-on (VTON) and mask-free virtual try-off (VTOFF) via dual-path conditioning: latent concatenation preserves structure, while pixel-level semantic and spatial modules refine details, fused by an extended attention block. A two-stage training strategy bridges mask asymmetry. Evaluations on VITON-HD and DressCode show TWGTM outperforms state-of-the-art methods across structure, texture, and perceptual metrics, with mutual task enhancement.

**Strengths:**

- First unified diffusion framework that jointly solves VTON and VTOFF in one model.
- Dual-space (latent + pixel) conditioning preserves global structure and fine texture simultaneously.
- Extended attention block enables seamless fusion of semantic and spatial features, boosting both tasks.
- Two-stage training eliminates the mask-dependency gap between masked VTON and mask-free VTOFF.
- Consistent SOTA scores on VITON-HD and DressCode with lower FID, LPIPS and DISTS.
- Mutual reinforcement: fine-tuning one task improves the other, confirming true bidirectional synergy.

**Weaknesses:**

- Color shifts remain on extreme-white/black garments due to lighting domain gaps.
- Accessories or specular highlights are occasionally misclassified as garment parts, creating artifacts.
- Heavy Transformer-based architecture raises inference cost versus single-task models.

**Questions:**

1. As shown in Variant 8 of Table 4, the proposed Q-Former structure—i.e., the Semantic Abstraction Module—contributes almost nothing to virtual try-on; once experimental error is taken into account, its impact is effectively zero. Is this module therefore redundant, and how can its effectiveness and influence be substantiated?

2. In the three-stage cascade of the Spatial Refinement Module, erroneous masks or blurry boundaries from earlier stages are progressively amplified. How do you counteract this error accumulation?

3. The authors employ a large number of colored arrows in Figure 2 whose meanings are never defined. What data flows do the different colors denote? For instance, what are the red versus purple arrows in the Spatial Refinement Module? Readers are left dizzy trying to decode them.

4. What do $\mathcal{L} {dice}$ and $\mathcal{L}_{bce}$ in Eq. (15) stand for, how are they implemented, and why are they never explicitly defined—even in plain text? I can only guess that L_bce denotes the standard binary-cross-entropy loss.

5. In the Extended Attention Block, why not use the LoRA block in the subsequent cross-attention to fuse FSAM and FSRM? After all, the self-attention output still carries F_in via the residual connection, so the whole upper cross-attention + ZLR branch could be replaced by that LoRA. Is the module therefore redundant?

6. The equations in the manuscript are hard to follow because of the abbreviations: e.g.,

- Eq. (10) and Eq. (11) could be merged into a single statement.
- The real space $\mathbb{R}$ should be written as $\mathbb{R}$, not the plain letter $R$ – the current typesetting is non-standard.
- In line 194, [.,.] is said to denote spatial concatenation along width, yet at the end of the paragraph it suddenly means channel-wise concatenation, and in Eq. (4) it appears to stand for yet another operation.
- H and W are first defined as image height and width, but on line 259 they suddenly denote “spatial resolution”; why not choose fresh symbols? This is needlessly confusing.
- Likewise, the letter L is reused for two different quantities in Eq. (14) and Eq. (16).

None of these symbols are consistently reflected in Fig. 2, forcing readers to guess their meanings.

After reviewing the entire manuscript, a **major revision** is strongly recommended. Good presentation is required to let valuable work shine.

---

### Official Review · Reviewer_L7Cr · 2025-11-02

**Soundness:** 3
**Presentation:** 2
**Contribution:** 3
**Rating:** 4
**Confidence:** 4

**Summary:**

This paper considers a model which can perform both virtual try on and virtual try-off. The proposed approach adopts the setup similar to CatVTON where the model image and the garment image are spatially concatenated to create a target image h. These are then concatenated channel-wise with: i) spatially concatenated masked model image and the original garment image and (ii) spatially concatenated mask of inpainting region and an all-ones mask. This work extends this setup for virtual try-off by reversing the order of spatial concatenation. This means that if we want to perform virtual try-off, the input is: i) spatial concatenation of garment image and model image, ii) mask of garment image and the original model image and iii) mask for inpainting region and all-ones mask. The garment mask is usually not available, so the model is trained in two phases: in the first phase the mask is produced from the garment image, while in the second, the input is just a rectangular mask. This method thus enables to use a single model for both VTON and VTOFF, just by reversing the order of images.

The model itself is further conditioned on image features extracted by a a CLIP encoder and hierarchical features extracted by a swin transformer. Encoded simple textual descriptions describing the type of garment (e.g. upper body) are further used as guidance. These various condititionings are combined through the use of  QFormer (combines CLIP features with text features) and Taskformer. The diffusion network uses an Extended attention block that enables the inclusion of all the various conditioning features.

**Strengths:**

[S1] The idea of training a single network for both Virtual Try-Off and Virtual Try-On is good, and experiments confirm that there is a strong relationship between the two tasks.

[S2] I also like the use of the CatVTOn setup, and the simplicity of swapping the ordering the spatial concatentation of model and garment image to achieve the multi-task capability.

**Weaknesses:**

[W1] The presentation of the work is poor, and important context about the method seems to be missing. See questions.

[W2] The setup is quite complex with various conditioning features extracted by different, an probably computationally intensive, networks. These choices are ablated, but the result that more capacity improves results is not that interesting. It would be more interesting to ablate the proper input choices (e.g. if the spatial concatenation needs to be reversed, or if all that can change is masking).

**Questions:**

[Q1] How exactly was the comparison between feature spaces for VTON and VTOFF done since the two networks are somewhat different?

[Q2] How were the masks for the garments extracted? Line 298 suggest that it is predicted by the model, but that does not make much sense for an untrained model. Overall, it is not clear how the model predicts the mask. Is it also diffused? Are the morphological operation in stage 2 explicitly applied, because that is not really expanded upon (line 305).

[Q3] Figure 2 would benefit from the inclusion of feature names form the Equations to more easily follow the flow of information. Also a figure that would clarify equation 15 (the mask loss) would be useful,

[Q4] The ablations in Table 4 are poorly organized and also quite haphazard. Referring to them just as variants further decreases readablity. Some are testing the individual architectural solutions, while others are testing the relations between the two task. Some are testing design choices not discussed in the main paper (e.g. ip-adapter, Mask2bBbox).

---

### Meta-Review · Area_Chair_eH4t · 2026-01-06

**Summary:**

The reviewers raise concerns about both clarity and content.  The presentation is poor and omits important methodological context, leaving key elements such as the role of the reference image insufficiently explained. The proposed approach is seen as overly complex, relying on multiple computationally expensive conditioning networks, while the ablation studies are considered uninformative, largely demonstrating that increased model capacity yields better performance rather than providing insight into critical design choices such as input configurations, spatial concatenation, or masking strategies. Core components, including the Spatial Refinement Module, lack clear motivation and justification, and the Semantic Abstraction Module appears insufficiently novel, closely resembling existing methods such as IP-Adapter. Performance improvements are marginal, and the heavy Transformer-based architecture raises concerns about inference cost compared to simpler single-task models, with parameter similarity claims (e.g., Figure 1a) insufficiently substantiated due to missing baselines. Qualitative results further reveal unresolved robustness issues, including color shifts under extreme lighting conditions and artifacts caused by misclassification of accessories or specular highlights, calling into question the practical impact of the method.

**Reviewer Concerns:**

There is no rebuttal provided by the authors

**Reviewer Scores:**

Given that there is no rebuttal, it is likely the reviewers would have kept their original scores.

---

### Decision · Program_Chairs · 2026-01-26

Reject